# *CYP2C9* Polymorphism Influence in PK/PD Model of Naproxen and 6-O-Desmethylnaproxen in Oral Fluid

**DOI:** 10.3390/metabo12111106

**Published:** 2022-11-13

**Authors:** Gabriela Moraes Oliveira, Thiago José Dionísio, Viviane Silva Siqueira-Sandrin, Leticia Alves de Lima Ferrari, Bruna Bolani, Viviane Aparecida Parisi, Nelson Leonel Del Hierro Polanco, Bella Luna Colombini-Ishikiriama, Flávio Augusto Cardoso Faria, Carlos Ferreira Santos, Adriana Maria Calvo

**Affiliations:** Department of Biological Sciences, Bauru School of Dentistry, University of São Paulo, Bauru 17012-901, São Paulo, Brazil

**Keywords:** pharmacogenetic, naproxen, CYP2C9, oral fluid

## Abstract

Polymorphisms in *CYP2C9* can significantly interfere with the pharmacokinetic (PK) and pharmacodynamic (PD) parameters of nonsteroidal anti-inflammatory drugs (NSAIDs), including naproxen. The present research aimed to study the PK/PD parameters of naproxen and its metabolite, 6-O-desmethylnaproxen, associated with allelic variations of *CYP2C9*. In our study, a rapid, selective, and sensitive Liquid Chromatography-Tandem Mass Spectrometry (LC-MS/MS) method was developed and validated for the determination of naproxen and its main metabolite, 6-O-desmethylnaproxen, in oral fluid. Naproxen and its main metabolite were separated using a Shim-Pack XR-ODS 75L × 2.0 column and C18 pre-column at 40 °C using a mixture of methanol and 10 mM ammonium acetate (70:30, *v*/*v*), with an injection flow of 0.3 mL/min. The total analytical run time was 3 min. The volunteers, previously genotyped for *CYP2C9* (16 ancestral—*CYP2C9 *1* and 12 with the presence of polymorphism—*CYP2C9 *2 or *3*), had their oral fluids collected sequentially before and after taking a naproxen tablet (500 mg) at the following times: 0.25, 0.5, 0.75, 1, 1.5, 2, 3, 4, 5, 6 8, 11, 24, 48, 72 and 96 h. Significant differences in the PK parameters (* *p* < 0.05) of naproxen in the oral fluid were: Vd/F (L): 98.86 (55.58–322.07) and 380.22 (261.84–1097.99); Kel (1/h): 0.84 (0.69–1.34) and 1.86 (1.09–4.06), in ancestral and mutated *CYP2C9 *2* and/or **3*, respectively. For 6-O-desmethylnaproxen, no PK parameters were significantly different between groups. The analysis of prostaglandin E2 (PGE_2_) proved to be effective and sensitive for PD parameters analysis and showed higher levels in the mutated group (*p* < 0.05). Both naproxen and its main metabolite, 6-O-desmethylnaproxen, and PGE_2_ in oral fluid can be effectively quantified using LC-MS/MS after a 500 mg oral dose of naproxen. Our method proved to be effective and sensitive to determine the lower limit of quantification of naproxen and its metabolite, 6-O-desmethylnaproxen, in oral fluid (2.4 ng/mL). All validation data, such as accuracy, precision, and repeatability intra- and inter-assay, were less than 15%. Allelic variations of *CYP2C9* may be considered relevant in the PK of naproxen and its main metabolite, 6-O-desmethylnaproxen.

## 1. Introduction

While polymorphic enzymes that metabolize and transport drugs can affect pharmacokinetics (PK), polymorphic variations and disease-related pathways can influence the pharmacodynamic action (PD) of drugs. Allelic variants of cytochrome P450 (CYP) C9 (*CYP2C9*) are potentially linked to an increase in adverse events induced by non-steroidal anti-inflammatory drugs (NSAIDs), such as increased gastrointestinal bleeding, ulcers, delays in drug excretion, and, consequently, greater toxicity [1,2,3,4].

After a usual dose of NSAID, variations in the toxicity and inefficiency of the drug can be observed, depending on the polymorphism of the NSAID considered and the presence or absence of active metabolites. Naproxen sodium is metabolized by the liver enzyme CYP2C9 to form 6-O-desmethylnaproxen. Single nucleotide polymorphisms (SNPs) in *CYP2C9*, especially *CYP2C9 *3* and **2*, could modify some PK parameters [5,6,7].

The real clinical consequences of the pharmacogenetics of NSAIDs are beginning to be understood as a preventive strategy in the management of inflammatory signs, such as pain, edema, and trismus. The selection of the best NSAID, at the best dose, in relation to a patient’s specific genotype is rarely considered. In this sense, the “right drug” for the “right patient” is still far from being prescribed, motivating further studies in this area.

At this time, the priority of research that investigates most of the polymorphisms should be to demonstrate the contribution of the genotype to the phenotype and then to examine the impact and its influence on the organism and the effects of NSAIDs, and the consequences of this variability in clinical outcomes. Several controlled studies still need to be carried out so that we can integrate pharmacogenetics into personalized prescriptions to improve the control of our patients’ inflammatory signs [8].

More recently, some researchers have described a new mechanistic approach to predict drug PK [9,10,11], called Physiologically Based Pharmacokinetics (PBPK). Studies using this model have shown that PBPK is useful for guiding dose adjustment in various clinical settings, such as the effects of genetic polymorphisms [11,12,13,14,15]. Some PBPK models for NSAIDs have been established in different *CYP2C9* genotypes [10,16,17].

However, the PK and PD information are solid scientific foundations of current pharmacotherapy and the data obtained from this information are essential for personalized prescribing studies based on pharmacogenetic studies. In this sense, the PK/PD model consists of linking PK and PD to establish and evaluate dose-concentration-response relationships and then describe and predict the time-effect courses arising from a drug dose [8,18].

For our study model of inflammation, some physiological options are well established in the study of the PK/PD model [19]. The synthesis of prostaglandin E2 (PGE_2_) is modulated by the cyclooxygenase-2 enzyme (COX-2). Changes in PGE_2_ can be used to quantify COX-2 inhibition after NSAID administration [20], which is our main investigation target for the PK and PD relationships of COX-2 inhibition according to the NSAID studied and the relationship with the *CYP2C9* polymorphism of the volunteers [21].

The use of oral fluid to carry out PK/PD studies with various drugs has been shown to be quite effective, ethical, and practical, as such studies require several collections after the ingestion of the drug under study [22,23,24]. Previous studies show that salivary concentrations of naproxen in healthy subjects treated with a single dose (550 mg) have concentrations of approximately 60% of the serum concentrations found in plasma (with Cmax of 24 mg/L, Tmax of 4 h and tl/2 24-h elimination), giving us confidence that the levels found in our study are valid for the proposed calculations and comparisons [25,26]. Its analysis in Liquid Chromatography-Tandem Mass Spectrometry (LC-MS/MS) allows the detection of very small fractions of the drug and also of its metabolites, especially when considering drugs as bound to proteins as NSAIDs [24].

Furthermore, the use of a recent extraction methodology, Microextraction by Packed Sorbent (MEPS), has been shown to be effective in this matrix for PK/PD studies [20,27]. MEPS technology consists of Solid-Phase Extraction (SPE) miniaturization with a compacted bed (1–2 mg). With this, it is possible to obtain a cleaner matrix, with less interference and specific analytes [20,28,29].

Thus, the present research aimed to study the PK/PD parameters of naproxen and its metabolite, 6-O-desmethylnaproxen, associated with allelic variations of *CYP2C9*.

## 2. Materials and Methods

This study was approved by the Research Ethics Committee of the Bauru School of Dentistry/University of São Paulo (CAAE 92312318.4.0000.5417) and is registered in Re-BEC (Brazilian Register of Clinical Trials RBR-38jcm9). All volunteers participating in this research signed the consent form after being fully informed about the study content and procedures.

### 2.1. Chemicals and Reagents

Naproxen (C_14_H_14_O_3_), 6-O-desmethylnaproxen (C_13_H_12_O_3_), prostaglandin E_2_ (C_20_H_32_O_5_-PGE_2_), and piroxicam (C15H13N3O4S) were purchased from Sigma-Aldrich (São Paulo, Brazil). Methanol, ammonium acetate, and other chromatographic grade chemicals used in the tests were purchased from Merck (Hohebrunn, Germany). During all experiments, water from a Milli-Q Plus purification system was used (Millipore, Belford, MA, USA).

### 2.2. Volunteers Selection and Sample Collection for PK Analysis

Twenty-eight volunteers who were previously genotyped for *CYP2C9* [23] were selected for this study. Oral fluids of all the participants were collected sequentially before and after taking a naproxen sodium tablet (500 mg) at the following times: 0.25, 0.5, 0.75, 1, 1.5, 2, 3, 4, 5, 6 8, 11, 24, 48, 72 and 96 h [22]. All samples were centrifuged at the same time for 10 min (2500 rpm) and the supernatant was stored at −20 °C until analysis.

The quantified pharmacokinetic parameters of naproxen and its main metabolite, 6-O-desmethylnaproxen, were estimated in oral fluid for the area under the curve from zero to the last quantifiable time AUC0-t, predicted total clearance (Cl/f), distribution volume (Vd/F), drug elimination half-life (t1/2), elimination constant (Kel), Time (Tmax) and value of the maximum observed concentration estimated (Cmax), based on concentrations obtained experimentally using a noncompartmental model with first-order elimination. These PK analyses were performed using Phoenix WinNonlin Software (version 8.1) (Certara L.p., Princeton, NJ, USA).

### 2.3. Genotyping CYP2C9

28 volunteers who were previously genotyped for *CYP2C9* [23], 16 of which were ancestral (*CYP2C9 *1/*1*) and 12 with the presence of polymorphism (*CYP2C9 *2* or **3*) were selected for this study. The distribution of the volunteers (*n* = 28) in relation to *CYP2C9* genotype polymorphisms is described in the results section.

The volunteers were instructed to collect 4 mL of unstimulated oral fluid using a provided sterile polypropylene collection tube. After, the oral fluid was divided into 2 microcentrifuge tubes. The microcentrifuge tube was immediately stored in a freezer at –20 °C until the genomic DNA was extracted. Then, a DNA Extract All Reagents Kit (catalog number 4402616, Applied Biosystems^®^, Foster City, CA, USA) was used for extracting the DNA from the oral fluid.

For the detection of the allelic variations of *CYP2C9*, TaqMan^®^ GTXpress™ Master Mix was used. DNA was analyzed using a Viia 7 Real-Time PCR (Applied Biosystems^®^) system. Each reaction was performed in duplicate. The manufactured assays were used and validated by Applied Biosystems^®^ with the following catalog numbers; In this study, primers and probes corresponded to *CYP2C9 *2* rs1799853 (430C → T) and *CYP2C9 *3* rs1057910 (1075A → C) (catalog numbers C_25625805_10 and C_27104892_10, respectively). DNA samples were quantified and qualified using TaqMan^®^ Genotyping Master Mix (catalog number: 4381656; Applied Biosystems^®^) [23,30].

### 2.4. Sample Preparation and Extraction Methodology

For all the oral fluid and calibration curves, MEPS was chosen as the extraction method for this research. MEPS consists of the miniaturization of conventional SPE with a conditioned needle incorporated directly into an injector syringe (SGE Analytical Science, Australia). This method improves drug isolation, targets analyte concentration, and reduces matrix effects [31,32]. This method has been well-reported as an alternative to other extraction methodologies in different matrixes [33], including oral fluid samples [9,20,29,34].

This extraction methodology basically consists of the steps of conditioning, sample, washing, elution, injection, and cleaning. The steps of the extraction procedure are listed in Table 1.

### 2.5. LC-MS/MS

Concentrations of naproxen, its main metabolic, 6-O-desmethylnaproxen, and PGE_2_ were analyzed using an 8040 Triple Quadrupole Mass Spectrometer (Shimadzu, Kyoto, Japan). Piroxicam was used as internal standard (IS). Drug characterization was performed using LC-MS/MS and separation was performed using a Shim-Pack XR-ODS 75 L × 2.0 column and C18 pre-column (Shimadzu, Kyoto, Japan) at 40 °C using a mixture of methanol and 10 mM ammonium acetate (70:30, *v*/*v*) with an injection flow of 0.3 mL/min. The total analytical run time was 3 min [22].

The Tandem Mass Spectrometry (MS/MS) conditions were optimized by the direct infusion of naproxen, 6-O-desmethylnaproxen, and PGE_2_ solutions at the concentration of 10 ng/mL and 1 ng/mL for IS solution. The multiple-reaction-monitoring (MRM) mode was used for quantification by monitoring the transitions that emerged during the analyses. The electrospray ionization (ESI) source was operated in MRM mode. The protonated ion [M + H] + and the respective production were monitored in the transitions of *m*/*z* 331.9 → 95.1; 121.05 for piroxicam IS. The negative mode was used to monitor precursors and products: *m*/*z* 228.9 → 169.25; 170.3 for naproxen, m/z 214.9 → 171.25; 169.25 for 6-O-demethylnaproxen, and *m*/*z* 351.1 → 315.3; 271.4 for PGE_2_ in oral fluid. Data acquisition and sample quantification were performed using the LabSolutions software, version 5.97 (Shimadzu, Kyoto, Japan).

### 2.6. Preparation of Standard Solutions and Validation

Stock solutions of naproxen, its main metabolite, 6-O-desmethylnaproxen, PGE_2_, and IS (all 1 mg/mL methanol) were prepared. A dilution series from each stock solution (10 µg/mL) was used to construct standard curves. The solutions were stored in the dark at −20 °C and all the stages of the research were conducted under a sodium vapor lamp to avoid the photodecomposition of the drugs used in the study [22,23,24].

The calibration curve for the oral fluid was prepared using the following concentrations: 1250, 625, 312.5, 156.2, 78.1, 39.1, 19.5, 9.8, 4.9, and 2.4 ng/mL for naproxen, its main metabolite 6-O-desmethylnaproxen, and PGE_2_. All the polypropylene tubes were stored at −20 °C until use.

For greater reliability of the results obtained in the PK assays in oral fluid, the method of analysis of naproxen and its main metabolite, 6-O-desmethylnaproxen, were validated for essential parameters such as matrix effects, calibration curve, residual effect, selectivity, precision, accuracy, stability, and dilution according to the current recommendations of the FDA U.S. Guidance for Industry: Bioanalytical Method Validation [35].

### 2.7. Statistical Analysis

Quantitative data were descriptively organized and analyzed using Sigmaplot (version 14.0). The Shapiro-Wilk test was performed to check the normality of the data, followed by the un-paired *t*-test or Mann-Whitney test to compare naproxen, 6-O-desmethylnaproxen, and PGE_2_. The results are presented as median (interquartile range). The level of significance was set at 5%.

## 3. Results

Oral fluid concentrations of naproxen, 6-O-desmethylnaproxen, and PGE_2_ were analyzed from the oral fluid collected from the volunteers, before and after, up to 96 h after a single oral dose of naproxen (500 mg). The descriptive data of the twenty-eight volunteers are presented in Table 2.

### 3.1. Genotyping CYP2C9

The genomic DNA from all oral fluid was extracted and genotyped. The results obtained from the 28 volunteers are described in Table 3. Sixteen volunteers were ancestral homozygotes (*CYP2C9 *1/*1*). Twelve volunteers had one or both allelic variations evaluated in this study. Six volunteers showed polymorphisms for *CYP2C9 *2* (heterozygotes (*CYP2C9 *1/*2*) = 5 and homozygotes (*CYP2C9 *2/*2*) = 1). For *CYP2C9 *3*, six volunteers showed polymorphisms (three heterozygotes (*CYP2C9 *1/*3*) and three heterozygotes for both polymorphisms (*CYP2C9 *2/*3*).

### 3.2. Calibration Curve of Naproxen, 6-O-Desmethylnaproxen, and PGE_2_

For the analysis of naproxen, 6-O-desmethylnaproxen, and PGE_2_ concentrations in oral fluid, a calibration curve with known standard concentrations was used as reference for the analysis of the concentrations in the patient samples, obtaining r² = 0.989 and f(x) = 14,801.9 × x − 8502.29, r² = 0.994 and f(x) = 1903.97 × x + 13,960.0 and r² = 0.988 and f(x) = 540.783 × x + 447.083, respectively. All the data for the optimization of the analysis of the methods and conditions are available in the data repository (https://data.mendeley.com/datasets/kjmphc464z/1, accessed on 7 October 2022). Considering the stability of the curves, analyses of the samples were performed.

### 3.3. PK Analysis

For naproxen and 6-O-desmethylnaproxen in oral fluid (200 μL aliquots) LLOQ was 2.4 ng/mL, LQC—9.8 ng/mL, MQC—312.5 ng/mL, and HQC—625 ng/mL. DQC used 2500 ng/mL of naproxen and 6-O-desmethylnaproxen in oral fluid.

PGE_2_ was detected in most of the samples and used to perform PD assays in oral fluid after the taking of a naproxen tablet (500 mg) by the volunteer. In addition, it was essential to the PK/PD model of this study.

In Figure 1, the mean concentrations found at each time point after the oral administration of a naproxen tablet (500 mg) demonstrate the metabolism profiles of naproxen (a) and 6-O-desmethylnaproxen (b) over time for *CYP2C9* genotypes. The concentrations found in Figure 1a do not show a significant difference of naproxen (a) between the ancestral and mutated groups. For 6-O-desmethylnaproxen (b), there was a significant difference between the ancestral and mutated groups throughout the collection period.

Table 4 shows the PK parameters obtained from the concentrations found of the naproxen and its main metabolite, 6-O-desmethylnaproxen, in the analyzed oral fluid samples, separated between the ancestral (*CYP2C9 *1*) and mutated (*CYP2C9 *2* and/or **3*) volunteers. The parameters Vd/F (L) and Kel (1/h) showed a significant difference between ancestral volunteers and those with *CYP2C9* polymorphism for naproxen. For 6-O-desmethylnaproxen, no significant differences were found between the groups studied.

For a better visualization of the results, the mean found for the PK parameters in each of the subgroups of SNPs mutated in *CYP2C9* are described in the Table 5, below:

### 3.4. PGE_2_ Analysis

The mean of the quantified concentrations of PGE_2_ over time is shown in Figure 2. There was a clear difference in the mean of the concentrations of PGE_2_ in the ancestral and mutated groups, with a significant difference when comparing the two groups. However, in the individual statistical analysis of each time analyzed, only the time of 2 h showed a significant difference (* *p* < 0.05).

### 3.5. Analytical Validation

The matrix effect of all oral fluid (*n* = 6) was absent (Table 6) for both naproxen and its main metabolite, 6-O-desmethylnaproxen. The coefficient of variation of the IS normalized was less than 15% for each analyte.

Analytical validation parameters for the methods of naproxen and 6-O-desmethylnaproxen in human oral fluid are presented in Table 6. In brief, linearity was 2.4 to 1250 ng/mL, r^2^ = 0.989 for naproxen, and 2.4 to 1250 ng/mL, r^2^ = 0.994 for its main metabolite, 6-O-desmethylnaproxen. The determination of precision and accuracy obtained showed a coefficient of variation <15%, which indicates an analysis with accuracy and reproducibility. The intra- and inter-assay precision and accuracy represented by the coefficient of variation and the relative error, respectively, were <15% for naproxen and its main metabolite, 6-O-desmethylnaproxen, in the oral fluid.

For stability tests, three freezes (−70 °C)/thaw (23 °C) cycles were performed. The oral fluid sample was kept stable after 12 h at 23 °C and the post-processing of the samples was kept in the autoinjector at 4 °C for 12 h.

All parameters were evaluated using freshly prepared calibration curves. Stability is accepted when the deviation from the nominal value is equal to or less than ±15%.

The quality control to achieve dilution integrity of naproxen and its main metabolite, 6-O-desmethylnaproxen, had a coefficient of variations <15%. The analytical validation of PGE_2_ was performed and published in a recent article by our research group [20,22].

## 4. Discussion

Allelic variations of *CYP2C9* are routinely related to increased adverse effects of NSAIDs [1,2,3,4]. An NSAID extensively metabolized by the liver enzyme CYP2C9 is naproxen, the object of study in this research. Naproxen sodium is metabolized by this enzyme to its main metabolite, 6-O-desmethylnaproxen. In the current literature, there are insufficient studies that report the influence of allelic variations in *CYP2C9* on the metabolism and efficacy of this drug [6].

The present research aimed to study the PK/PD model of naproxen and its metabolite, 6-O-desmethylnaproxen, associated with allelic variations of *CYP2C9*, using oral fluid as a source of material for study. Previous studies by our group [24] showed a good relationship between plasma levels in relation to the levels found in the oral fluid of other NSAIDs, further motivating this research.

Volunteers with allelic variations of *CYP2C9*, such as *CYP2C9 *1/*2, *2/*2, *1/*3*, and **2/*3*, were found in this research. Studies report that these allelic variations can influence the type of metabolism that the drug will undergo [6,36]. Thus, different predicted phenotypes within a population could be classified as “poor metabolizer”, “intermediate metabolizer”, “extensive metabolizer” or “ultra-rapid metabolizer” [37,38].

It is possible to observe that the general mean of naproxen concentrations over time showed no statistical difference between mutated and ancestral volunteers. Despite these data, when we relate them to the PGE_2_ concentrations found, they demonstrate that the lowest PGE_2_ concentration over time was statistically significant (** *p* < 0.001) in volunteers with allelic variants (*CYP2C9* mutated **2* and/or **3*), specifically, at 2 h, when the drug starts to show the most evident anti-inflammatory effects (**p* < 0.05). There is also a gradual return to the normal levels of each individual around 8 h; taking into account that only one naproxen tablet (500 mg) was administered in this research and its use can be up to 12 h, an increase can be expected of PGE_2_ levels close to this period.

Considering the general average of the concentrations found for 6-O-desmethylnaproxen, it was possible to observe that volunteers with genetic polymorphisms for *CYP2C9* had a prolonged time of the metabolite in the oral fluid, when compared to ancestral volunteers. Possibly, this may have occurred because mutated volunteers could be described as poor or moderate metabolizers. Studies report that *CYP2C9* polymorphisms may be related to and interfere with the formation of the 6-O-desmethylnaproxen metabolite. [5,6,39,40].

Regarding the PK parameters of naproxen, the volunteers mutated to *CYP2C9* had higher values of elimination constant (Kel) and volume of distribution (Vd/F) than the ancestral group (* *p* < 0.05). Thus, naproxen acts by inhibiting COX-2 and consequently reduces the production of PGE_2_, as it was demonstrated in the mutated volunteers. However, with these results, it was possible to observe that the circulation time of the drug in the body and in the liver can increase. As a result, drug metabolites take longer to produce and appear in high concentrations in the final hours, concomitant with our results.

In our research group, recent studies [20,22,23,24,41] showed promising results of bio-equivalence in oral fluid for pharmacokinetic studies with NSAIDs. MEPS technology proved to be effective and sensitive for the extraction of analytes detected and quantified in this research. The method was developed and validated according to the parameters of the FDA Guide for Industry (USA) with the coefficients of variation and standard errors of relative precision less than 15%.

As already established, PGE_2_ can be investigated for the PK/PD ratios of COX-2 inhibition [20,21]. In our study, it was possible to evaluate the quantification of PGE_2_ concomitantly with the PK parameters of naproxen and the correlation with allelic variations of *CYP2C9*, demonstrating that mutated individuals may have lower levels of PGE_2_ over time, together with longer permanence of naproxen in the body, which may lead to possible side effects, when we think of patients who need to make chronic use of this class of drugs. Being able to move towards a personalization of the prescription in cases that are possible and beneficial for the patient.

Studies that evaluated the interference of genetic polymorphisms in PK/PD models indicate that the future of personalized medicine is getting closer. New methods, such as the PB/PK model, are innovative and have been shown to be effective. In recent published articles [9,11,12,13,15,18], the PBPK model related to the genetic polymorphism of *CYP2C9* established and described the pharmacokinetics of piroxicam in different *CYP2C9* genotypes. The PBPK model can be used as an alternative to predict PK in different clinical conditions [10,11,14,16,17,19].

## 5. Conclusions

Considering the genetic analysis of this study, allelic variations in *CYP2C9* can be considered relevant in the pharmacokinetics of naproxen and its main metabolite, 6-O-desmethylnaproxen, as well as greater inhibition of COX-2 and consequently lower PGE_2_ levels over time for mutated individuals.

Both naproxen and 6-O-desmethylnaproxen can be effectively quantified in oral fluid using LC-MS/MS after a 500 mg oral dose of naproxen in healthy volunteers. The methodology developed was fast, sensitive, precise, and selective for each drug and allowed the analysis of its PK parameters. The methodology also allowed the detection and quantification of PGE_2_ concentrations, enabling the PK/PD model in oral fluid.

## Figures and Tables

**Figure 1 metabolites-12-01106-f001:**
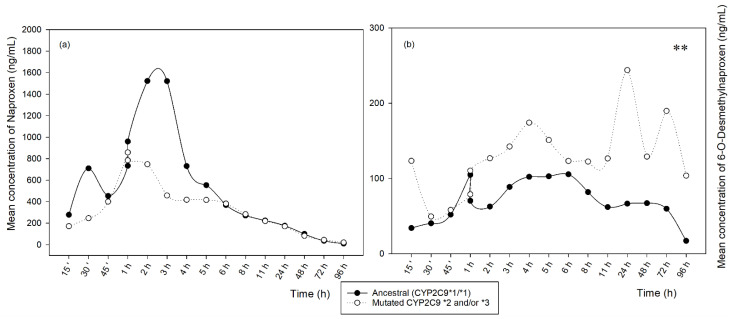
Mean concentrations of naproxen (**a**) and 6-O-desmethylnaproxen (**b**) in relation to the time analyzed for oral fluid samples from ancestral and mutant patients. (Statistically significant difference ** *p* < 0.001).

**Figure 2 metabolites-12-01106-f002:**
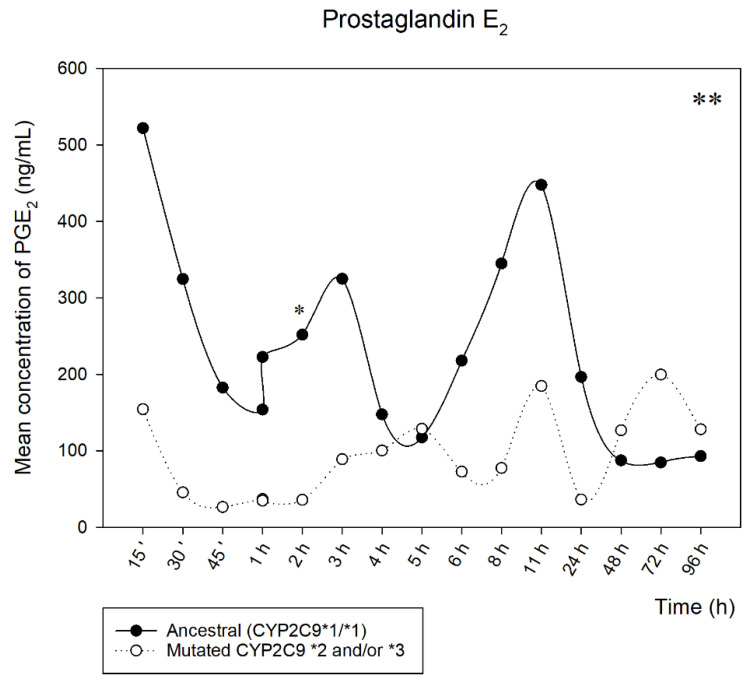
Oral fluid concentrations of PGE_2_ in volunteers considered ancestral and mutated for *CYP2C9*, in relation to the time analyzed, after 500 mg of naproxen. (Statistically significant difference ** *p* < 0.001; * *p* < 0.05).

**Table 1 metabolites-12-01106-t001:** Extraction method steps using MEPS.

MEPS Procedure Step	The Optimized Parameters
Conditioning	Methanol (100 μL) and water (2 × 50 μL) consecutively, before first use
Extraction	Draw-ejected, 100 μL of oral fluid
Washing	Milli-Q water (50 μL)
Elution	10 mM ammonium acetate + methanol (80:20, *v*/*v*), 1 × 100 μL
Injection	Dispensed into the vial and injected directly into the LC-MS/MS 8040
Washing solution	(10 × 100 µL—Methanol) between each volunteer sample

**Table 2 metabolites-12-01106-t002:** Descriptive data of the sample (*n* = 28).

Female *n* (%)	Male *n* (%)	Age—Years (Mean ± SD)	Body Weight—kg (Mean ± SD)	Height—m (Mean ± SD)
20 (71.4)	8 (28.6)	30.39 ± 9.94	70.92 ± 19.65	1.65 ± 0.11

**Table 3 metabolites-12-01106-t003:** Distribution of the volunteers (*n* = 28) in relation to *CYP2C9* genotype.

	Ancestral	Mutated *CYP2C9 *2* and/or **3* (*n* = 12)			
	*CYP2C9* **1/*1*	*CYP2C9* **1/*2*	*CYP2C9* **2/*2*	*CYP2C9* **1/*3*	*CYP2C9* **2/*3*
Volunteers	16	5	1	3	3

**Table 4 metabolites-12-01106-t004:** Pharmacokinetic parameters of naproxen and its main metabolite, 6-O-desmethylnaproxen, for *CYP2C9* genotypes.

PK Parameters	Naproxen		*p*-Value
	Ancestral *CYP2C9 *1* (*n* = 16)	Mutated *CYP2C9 *2* and/or **3* (*n* = 12)	
No Normal Distribuition	Median (Q1–Q3)		
Tmax (h)	1.96 (1.50–2.30)	1.45 (0.91–1.99)	0.144
AUC0-t (h × ng/mL)	5700.37 (2966.69–11,619.93)	4613.98 (1554.67–7891.63)	0.341
Cl/F (L/h)	96.53 (47.33–185.80)	130.33 (69.69–356.14)	0.341
Vd/F (L)	98.86 (55.58–322.07)	380.22 (261.84–1097.99) *	0.039
Cmax (ng/mL)	1980.79 (659.13–3610.94)	718.33 (367.02–1373.52)	0.131
Kel (1/h)	0.84 (0.69–1.34)	1.86 (1.09–4.06) *	0.039
T1_/2_ (h)	0.83 (0.372–2.086)	2.77 (0.77–4.86)	0.186
**PK Parameters**	**6-O-Desmethylnaproxen**		** *p* ** **-Value**
**No Normal Distribuition**	**Median (Q1–Q3)**		
Tmax (h)	5.21 (3.65–21.78)	6.78 (2.83–19.81)	0.871
AUC0-t (h × ng/mL)	2235.45 (864.31–7669.62)	1809.93 (900.53–11,017.08)	0.908
Cl/F (L/h)	261.41 (72.17–636.64)	308.03(59.82–614.67)	0.908
Vd/F (L)	1573.55 (1077.81–2373.21)	1384.99 (783.79–2918.91)	0.626
Cmax (ng/mL)	128.84 (85.45–188.68)	152.88 (72.06–258.38)	0.626
Kel (1/h)	0.21 (0.04–0.31)	0.22 (0.052–0.39)	0.472
T1_/2_ (h)	3.31 (2.22–15.10)	4.13 (1.76–13.72)	0.472

Tmax e Cmax: time and value of the maximum observed concentration, respectively; AUC0-t: area under concentration versus time curve from the first observed concentration to the last one; Vd/F: estimated volume of distribution in total AUC; Vd: CL/Kel; Clt/F: full clearance; CL = dose/AUC0-t; Kel: elimination rate constant estimated from the regression line representing the terminal phase of the concentration-time profile; T1/2: terminal half-life of the drug. (Statistically significant difference * *p* < 0.05.

**Table 5 metabolites-12-01106-t005:** Mean values of PK parameters of SNP subgroups mutated in *CYP2C9*.

	Naproxen (Mean ± SD)			
PK Parameters	Mutated *CYP2C9 *1/*2* (*n* = 5)	Mutated *CYP2C9 *1/*3* (*n* = 3)	Mutated *CYP2C9 *2/*3* (*n* = 3)	Mutated *CYP2C9 *2/*2* (*n* = 1)
Tmax (h)	1.56 ± 1.05	1.44 ± 0.34	1.85 ± 0.69	0.820
AUC0-t (h × ng/mL)	14,949.15 ± 20,058.54	3474.07 ± 3869.14	5511.24 ± 3213.44	880.49
Cl/F (L/h)	136.47 ± 144.79	374.83 ± 343.48	149.29 ± 128.26	624.65
Vd/F (L)	609.51 ± 622.08	563.75 ± 225.70	619.38 ± 547.85	391.95
Cmax (ng/mL)	1782.51 ± 2112.46	617.26 ± 364.61	1012.28 ± 602.66	516.21
Kel (1/h)	11.34 ± 20.61	2.51 ± 2.61	1.93 ± 0.24	1.59
T1_/2_ (h)	18.06 ± 31.98	2.23 ± 2.48	2.84 ± 0.19	0.43
	**6-O-desmethylnaproxen (Mean ± SD)**			
Tmax (h)	16.42 ± 15.09	11.83 ± 9.66	3.23 ± 1.25	13.8
AUC0-t (h × ng/mL)	13,499.10 ± 20,453.47	6511.06 ± 5908.19	981.38 ± 899.89	3712.5
Cl/F (L/h)	325.08 ± 288.51	166.32 ± 157.77	885.80 ± 533.50	148.1
Vd/F (L)	1734.15 ± 1749.44	1676.27 ± 1887.09	1934.06 ± 1202.80	2037.0
Cmax (ng/mL)	227.59 ± 185.87	327.01 ± 340.25	140.94 ± 93.86	99.3
Kel (1/h)	0.22 ± 0.22	0.18 ± 0.19	0.45 ± 0.17	0.1
T1_/2_ (h)	10.95 ± 10.91	8.20 ± 6.69	1.65 ± 0.53	9.5

**Table 6 metabolites-12-01106-t006:** Validation of Naproxen and 6-O-desmethylnaproxen quantification methods in oral fluid.

	Naproxen		6-O-Desmethylnaproxen	
	Concentration (ng/mL)	IS Normalized Matrix Factor (CV)	Concentration (ng/mL)	IS Normalized Matrix Factor (CV)
**Oral fluid** (6 samples)				
LQC	9.8	4.26	9.8	8.34
HQC	625	9.40	625	5.21
**Linearity**				
*r* ^2^	0.989		0.994	
Equation of the line	f(x) = 14801.9 × x−8502.29		f(x) = 1903.97 × x + 13960.0	
Limit of quantification (ng/mL)	2.4		2.4	
Precision (CV %; *n* = 10)	11.06		10.71	
Accuracy (%)	−5.05		−3.88	
**Precision (CV.%) and Accuracy (RE.%)**				
**Intra-assay (n = 5)**	**CV**	**RE**	**CV**	**RE**
LLOQ (2.4 ng/mL)	9.80	−2.78	8.45	−5.15
LQC (9.8 ng/mL)	9.44	5.33	7.92	0.32
MQC (312.5 ng/mL)	4.81	7.42	5.16	9.98
HQC (625 ng/mL)	5.57	6.91	5.39	8.44
DQC (2500 ng/mL; 1:5)	9.01	6.42	5.51	7.33
**Interassay (n = 8)**				
LLOQ (2.4 ng/mL)	8.75	−5.26	8.17	−1.89
LQC (9.8 ng/mL)	6.49	−4.84	9.23	−1.21
MQC (312.5 ng/mL)	6.02	4.56	8.59	6.68
HQC (625 ng/mL)	9.81	4.42	9.69	1.9
**Stabilities (*n* = 3)**				
Short-term stability (12 h at 23 °C)				
LQC (9.8 ng/mL)	5.09		1.12	
HQC (625 ng/mL)	−1.84		9.09	
Post-processing stability (12 h at 4 °C)				
LQC (9.8 ng/mL)	7.84		3.48	
HQC (625 ng/mL)	−2.33		−1.52	
Freeze/thaw cycle stability (−70 °C)				
LQC (9.8 ng/mL)	−10.72		0.68	
HQC (625 ng/mL)	−3.77		6.12	

IS: internal standard; CV: coefficient of variation [(standard deviation/mean) × 100]; r: linear correlation coefficient; RE: relative error = [(observed concentration−nominal concentration)/nominal concentration] × 100; LLOQ: lower limit of quantification; LQC: low quality control; MQC: medium quality control; HQC: high quality control; DQC: quality control for dilution integrity.

## Data Availability

Data available in a publicly accessible repository. The data presented in this study are openly available in Mendeley Data at [doi:10.17632/kjmphc464z.1], reference number [42].

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
