# Peer review of "CYP2C9 Polymorphism Influence in PK/PD Model of Naproxen and 6-O-Desmethylnaproxen in Oral Fluid"

_metabolites, 2022, doi:10.3390/metabo12111106_

Round 1

Reviewer 1 Report

The study examines the relationship between naproxen, 6-O-desmethyl-naproxen, and PGE2 in oral fluid  with allelic variations of CYP2C9 in 28 volunteers. The underlying hypothesis supporting the methodological approach is that the use of oral fluid to carry out this kind of studies is effective and practical in comparison to studies addressing blood/plasma levels of drugs and related metabolites.

However, to support the working hypothesis, preliminary data showing the degree of correlation between drug levels in oral fluid and blood/plasma would be necessary. This is even more necessary considering the results reported in the manuscript, which show extremely variable levels over time and which rise and fall several times. In addition, did any relationships between naproxen/6-O-desmethyl-naproxen levels and PGE2 levels occur over time? Indeed, while any relationship would strengthen the PK/PD model, absence of relationship would largely weaken the general meaning of the study.

Other issues include:

(*) why CYP2C9 *2 or *£ subjects were just pooled and not also shown in separate groups (as the authors themselves also did in other published work)?

(*) how Vd/F and CL/F were calculated? By use of which kind of values? Please detail.

Author Response

Response to Reviewer 1 Comments

  1. Point 1:The study examines the relationship between naproxen, 6-O-desmethyl-naproxen, and PGE2 in oral fluid with allelic variations of CYP2C9 in 28 volunteers. The underlying hypothesis supporting the methodological approach is that the use of oral fluid to carry out this kind of studies is effective and practical in comparison to studies addressing blood/plasma levels of drugs and related metabolites. However, to support the working hypothesis, preliminary data showing the degree of correlation between drug levels in oral fluid and blood/plasma would be necessary. This is even more necessary considering the results reported in the manuscript, which show extremely variable levels over time and which rise and fall several times. In addition, did any relationships between naproxen/6-O-desmethyl-naproxen levels and PGE2 levels occur over time? Indeed, while any relationship would strengthen the PK/PD model, absence of relationship would largely weaken the general meaning of the study.

Response 1: Thanks for the comment. Our study Calvo et al, 2016. (Quantification of Piroxicam and 5′-Hydroxypiroxicam in Human Plasma and Saliva Using Liquid Chromatography–Tandem Mass Spectrometry Following Oral Administration. J. Pharm. Biomed. Anal. 2016, 120, 212–220, doi :10.1016/j.jpba.2015.12.042) carried out such a comparison between both fluids, with very proportional results, where the whole idea of this project was initiated. And, we can say, that even in plasma samples, the levels can undergo such variations, when observed time to time. Hence the need for a high number of participants in all PK/PD studies.

As described in the text: "There was a clear difference in the mean of the concentrations of PGE2 in the ancestral and mutated groups, with a significant difference when comparing the two groups", taking this sentence into account and relating it to T1/2 (that, despite not being statistically significant) in mutated individuals to be almost 3 times higher than in the ancestors, we can infer the inhibition of COX-2 in such individuals was more effective, since the levels of PGE2 were significantly lower than in the ancestors over time.

  1. Point 2: (*) why CYP2C9 *2 or *£ subjects were just pooled and not also shown in separate groups (as the authors themselves also did in other published work)?

Response 2: Thanks for the comment, in other articles we received several suggestions to group the mutated individuals to obtain a larger number for comparison, since the search for such individuals is quite complicated by the prevalence in the population.

  1. Point 3: (*) how Vd/F and CL/F were calculated? By use of which kind of values? Please detail.

Response 3: Thanks for the comment. The WinNonlin software itself performs the calculations automatically, in short, the concentrations found at each moment for each volunteer are entered into the software and a non-compartmental model with first-order elimination is executed.

Reviewer 2 Report

1) Rewrite extraction methods..page 3 line 144 -157. 

2) page 4, line 192 - 197..should include under RESULTS

3) ancestral...is better to use "wild-type", this word is more common

4) No need to mention "Figure 2 etc" & "Table 2 etc" in DISCUSSION as you have mention it in the RESULTS

Author Response

Response to Reviewer 2 Comments

  1. Point 1: Rewrite extraction methods..page 3 line 144 -157. 

Response 1: Thanks for the comment. It was rewritten and described in Table 1

  1. Point 2: page 4, line 192 - 197..should include under RESULTS

Response 2: Thanks for the observation, it was replaced.

  1. Point 3: ..is better to use "wild-type", this word is more common

Response 3: Thanks for the comment. We did a literature search and found both descriptions, we opted for the "ancestral" in this paper.

  1. Point 4: No need to mention "Figure 2 etc" & "Table 2 etc" in DISCUSSION as you have mention it in the RESULTS

Response 4: Thanks for the comment, they have been removed.

Reviewer 3 Report

This manuscript is describing about CYP2C9 polymorphism influence in PK/PD model of naproxen and 6-O-desmethylnaproxen in oral fluid. The authors analyzed the metolites by susing  a rapid, selective, and sensitive Liquid Chromatography-Tandem Mass Spectrometry (LC-MS/MS).  They observed that both naproxen and its main metabolite, 6-O-desmethyl- naproxen, and PGE2 in oral fluid can be effectively quantified using LC-MS/MS after a 500 mg oral dose of naproxen, and concluded that allelic variations of CYP2C9 may be considered relevant in the PK of naproxen and its main metabolite, 6-O-desmethylnaproxen. This manuscript is recommended to be acccepable after minor revision.

- 6-O-desmethylnaproxen or 6-0-desmethylnaproxen ->6-O-desmethylnaproxen in entire manuscript.

- A space should be added between numerals and units. (ex, L147, or other 100uL -> 100 uL.)

- L252, Figure 1 -> Figure 2

- In Figures 1 and 2, 1h -> 1 h, 2h -> 2 h, ...........

Author Response

Response to Reviewer 3 Comments

This manuscript is describing about CYP2C9 polymorphism influence in PK/PD model of naproxen and 6-O-desmethylnaproxen in oral fluid. The authors analyzed the metolites by susing  a rapid, selective, and sensitive Liquid Chromatography-Tandem Mass Spectrometry (LC-MS/MS).  They observed that both naproxen and its main metabolite, 6-O-desmethyl- naproxen, and PGE2 in oral fluid can be effectively quantified using LC-MS/MS after a 500 mg oral dose of naproxen and concluded that allelic variations of CYP2C9 may be considered relevant in the PK of naproxen and its main metabolite, 6-O-desmethylnaproxen. This manuscript is recommended to be acccepable after minor revision.

  1. Point 1:- 6-O-desmethylnaproxen or 6-0-desmethylnaproxen ->6-O-desmethylnaproxen in entire manuscript.

Response 1: Thanks for the correction, the writing has been corrected throughout the text.

  1. Point 2: - A space should be added between numerals and units. (ex, L147, or other 100uL -> 100 uL.)

Response 2: Thanks for the observation, the values have been corrected.

  • Point 3: L252, Figure 1 -> Figure 2

Response 3: Thanks for the observation, the figure has been corrected.

 Point 4: In Figures 1 and 2, 1h -> 1 h, 2h -> 2 h, ...........

Response 4: Thanks for the observation, the values have been corrected.

Reviewer 4 Report

The article by Oliveira et al. has been reviewed. In the article, the authors have presented some interesting findings. However, a few more corrections are needed. 

  • The rationale of the project is not clear. Could you please elaborate more on why you conducted this study? Is it solely based on demonstrating the usefulness of the new analytical method developed?  
  • Many inconsistencies were noticed throughout the manuscript. Please correct for uniformity. E.g., 50-70 µl and 100µl, etc. 
  • Although naproxen T1/2 (h) is not statistically significant between mutated and ancestral volunteers, the change difference between the two groups is more than 3-fold. Could you please discuss more, illustrating previous studies? 
  • PEG2 level was found to be increased again after 8 h of drug administration. Is it a normal phenomenon after the administration of naproxen? 
  • Discussion is insufficient. Please compare your study results with previous studies and make some concrete conclusions.  
  • What is the future scope of this study? 

Author Response

Response to Reviewer 4 Comments

The article by Oliveira et al. has been reviewed. In the article, the authors have presented some interesting findings. However, a few more corrections are needed. 

  1. Point1: The rationale of the project is not clear. Could you please elaborate more on why you conducted this study? Is it solely based on demonstrating the usefulness of the new analytical method developed?  

Response 1: In addition to the development of the methodology, which you mentioned, the primary intention of this project is the possibility of using oral fluid to improve the living conditions of patients, especially those who make chronic use of drugs of this class, where adverse reactions could be minimized in the future. We add such an explanation in the discussion to make it clearer.

  1. Point 2: Many inconsistencies were noticed throughout the manuscript. Please correct for uniformity. E.g., 50-70 µl and 100µl, etc.

Response 2: Thanks for the observation, the values have been corrected (100 µl).

  1. Point 3: Although naproxen T1/2 (h) is not statistically significant between mutated and ancestral volunteers, the change difference between the two groups is more than 3-fold. Could you please discuss more, illustrating previous studies? 

Response 3: Thanks for the note. This was a surprise to us, but statistically, the variation that may have occurred between individuals may have masked this result. Indeed, individuals who are "poor or inter-mediate metabolizers" are expected to have a higher concentration of the drug for a longer time in the body, which our study also suggests, since PGE2 itself in these individuals remained at lower levels over time. and with significant differences, showing that COX-2 was inhibited for a longer time.

  1. Point 4: PEG2 level was found to be increased again after 8 h of drug administration. Is it a normal phenomenon after the administration of naproxen? 

Response 4: Thanks for questioning. Considering that only one naproxen tablet (500 mg) was administered and its use can be up to every 12 hours, the increase in PGE2 levels close to this period can be expected. Such information was added in the Discussion.

  1. Point 5: Discussion is insufficient. Please compare your study results with previous studies and make some concrete conclusions.  

Response 5: Thanks for the note, new paragraphs have been added to the discussion to make it more complete.

  1. Point 6: What is the future scope of this study? 

Response 6: We intend to expand the studies in pharmacogenetics, as well as the PK/PD studies, with saliva samples. Our current and previous results have shown to be a very rich and reliable source of material for such studies. We add such an explanation in the discussion to make it clearer.

Round 2

Reviewer 1 Report

In the previous review round, four points were raised, namely:

1. the need for data showing a relationship between between drug levels in oral fluid and blood/plasma;

2. the need to assess any relationships between naproxen/6-O-desmethyl-naproxen levels and PGE2 levels over time;

3. the request to show CYP2C9 *2 or *3 subjects not just as pooled but also in separate groups;

4. the request to detail how Vd/F and CL/F were calculated.

Apparently, none of these points was addressed in the present version of the submission. Point 1 in particular is essential, otherwise the whole study would lose any credibility (and of course piroxicam is not naproxen, data are needed for naproxen in the same way they were previously provided for piroxicam).

The other points can be very easily addressed, point 2 for example through a simple linear regression of naproxen/6-O-desmethyl-naproxen levels and PGE2 levels at each time point, point 3 by simply including separate groups together with pooled groups (and it will be possible to appreciate whether different SNPs have the same effects on naproxen PK), point 3 by detailing the model (it would be not acceptable to be forced to “believe” in not verifiable calculations automatically performed by a proprietary software).

Reviewer 4 Report

Answers from the authors are satisfactory. This article can be considered to be published in the Nutrienta.

Author Response

Thank you!